# Chemical Intolerance Is Associated with Autism Spectrum and Attention Deficit Disorders: A Five-Country Cross-National Replication Analysis

**DOI:** 10.3390/jox16010005

**Published:** 2026-01-01

**Authors:** Raymond F. Palmer, David Kattari

**Affiliations:** 1Department of Family and Community Medicine, University of Texas Health Science Center at San Antonio, San Antonio, TX 78229, USA; 2Kattari Consulting, Cayucos, CA 93430, USA; dkattari13@gmail.com

**Keywords:** chemical intolerance, multiple chemical sensitivity, autism, attention deficit hyperactive disorder

## Abstract

Background: Chemical Intolerance (CI), Autism Spectrum Disorder (ASD) and Attention Deficit Hyperactive Disorder (ADHD) are conditions with rising incidence rates not fully explained by greater awareness or changes in diagnostic practices. It is now generally accepted that the interaction between genetic and environmental exposures plays a role in all of these conditions. Prior studies show that these conditions co-occur. This study seeks to explore previous findings using an international sample. Methods: A five-country (*N* = 5000) stratified panel survey was used to assess self-reported CI in themselves, and ASD and ADHD in their children. A generalized linear model was used to estimate Odds Ratios. Age- and sex-adjusted logistic models used CI as a predictor of ASD and ADHD in separate models. Results: Compared to those classified as Low CI, High levels of CI were associated with greater Odds Ratios (OR) of reporting a child with ASD and ADHD in all countries except Japan. Italy, India, and the USA had over twice the OR of reporting a child with ASD. Mexico had over 1.9 times the OR. The results with ADHD are similar to the ASD results. Conclusions: The results of this study are consistent with two prior U.S. studies, showing an association between ASD and ADHD among women who have CI. However, cross-cultural comparisons, especially prevalence estimates for ASD and ADHD, cannot be interpreted as epidemiologic rates due to serious limitations of the survey methodology. No causal relationship should be inferred from this study.

## 1. Introduction

Chemical Intolerance (CI) is characterized by individuals presenting with adverse multi-system symptoms that are initiated either by a one-time high-dose exposure or persistent low-dose exposures to environmental toxicants [1,2]. New-onset intolerances are often triggered by subsequent exposures to structurally unrelated chemicals [3,4], foods [5,6], and/or drugs [7]. Symptoms may include combinations of fatigue, headache, mood changes, weakness, rash, dizziness, musculoskeletal pain, gastrointestinal disturbance, cognitive difficulties, and respiratory problems [1,2,8,9].

Different population-based surveys yield differing prevalence estimates depending on whether CI is clinically diagnosed (prevalence estimates between 0.5 and 6.5%) or self-reported (up to ~20+%) [10,11,12,13,14]. Explaining the differences in prevalence rates points to the diagnostic complexity which includes a lack of a universally accepted case definition and/or the use of different assessment tools [2,15].

The persistent lack of confirmed pathogenic mechanisms and varied non-specific, multi-system symptoms, and/or the wide variety of specific triggers to explain CI has historically led some health practitioners and researchers to accept a psychogenic theory of the condition, or at best, the more neutral descriptive term “medically unexplained symptoms” or “idiopathic environment-associated disease” [16]. There remains an ongoing debate in the literature about whether the symptoms of CI are purely physical (e.g., neurogenic inflammation) or influenced by psychogenic factors (e.g., classical conditioning or hypervigilance). These issues make clear diagnostic pathways difficult and contentious [2,16]. The theories and controversies surrounding CI are well-covered in Zucco and Doty [2]

Linking CI assessment to ongoing studies of neurodegenerative disease biology may eventually resolve the impasse over the psychogenic vs. biological explanations for CI [2]. Addressing the biological aspects of CI is highly relevant because (1) the prevalence of CI has been shown to be increasing globally [17,18,19,20], and (2) the broad social/economic burden is substantial (e.g., reduced workforce participation, disability and financial hardship, social isolation, high healthcare costs) [21,22].

### Autism Spectrum Disorder (ASD) and Attention Deficit Hyperactive Disorder (ADHD)

Both ASD and ADHD are common behaviorally defined neurodevelopmental disorders characterized by deficits in language, communication, sensory processing, and social function [23,24]. Prevalence estimates for ASD range from 1 in 30–44 U.S. births, with an estimated global prevalence of 1 per 100 children [25,26,27]. Worldwide, ADHD affects approximately 6% of youth and 2.5% of adults [28]. There is substantial overlap between autism and ADHD, with their co-occurrence estimated to be between 28% and 78% [29]. Other comorbidities include asthma, allergies, diabetes mellitus—immune and metabolic disorders [30,31,32,33,34,35,36].

Over the last 30 years, ASD and ADHD prevalence in the U.S. has undergone substantial increases, which cannot be fully explained by increased awareness, better access to healthcare, broadened diagnostic criteria, and/or better diagnostic practices [37,38]. ASD and ADHD share many biological irregularities with CI. Key mechanisms linking CI to ASD/ADHD involve the limbic and immune system (chronic inflammation) metabolic dysfunction through oxidative stress [39,40,41,42,43,44,45].

Study Purpose: In two different population surveys conducted in the United States, it has previously been reported that CI was associated with over three times the odds of having a child with either ASD or ADHD [46,47]. However, it should be noted that the inherent limitations of these self-reported cross-sectional studies preclude any inference of causality. Notwithstanding, since there is an increasing prevalence of CI, ASD, and ADHD globally, the purpose of this study is to explore this association using an international sample in order to test the universality vs. cultural contingency of the CI–ASD/ADHD link. Confirming this association in this sample would then justify future studies involving the investigation of several unknown potential biological mechanisms, including maternal vulnerabilities, shared endophenotypes, or perhaps as a proxy for environmental load.

## 2. Materials and Methods

Participants were recruited in 2020 from May 18th to the 20th through Dynata, a global, research-literate, multilingual data collection firm that provides recruitment services for academic and industry researchers (www.dynata.com) (accessed on 1 December 2025). Dynata adheres to the European Society for Opinion and Marketing Research code of conduct. This was an observational cross-sectional stratified panel survey consisting of five countries (the United States, India, Japan, Italy, and Mexico). Dynata performed survey translation, including back-translation, for each country.

Invitations to participate included emails, phone alerts, online banners, and community-site messaging, designed to reach individuals with diverse motivations for participating in research. Weighted randomization was used to assign surveys to participants. A review of the data was performed to ensure that answers were logical and not random responses, with additional logic checks built into the script to ensure participants could not continue if they tried to submit illogical answers. Overuse of item non-responses (e.g., ‘Don’t Know’) were identified and removed from the final data during quality checks.

Our sample was collected through the stratification of approximately equal numbers of participants (*N* = 1000) across three age bands: 18 to 34 years old, 35 to 54, and 55 and older, and by sex for approximately equal numbers of males and females. Respondents needed to be at least 18 years old. The study’s purpose was explained to participants who were informed that the survey would be anonymous. Consent was obtained online before the survey was administered. This research program was approved by the University of Texas Health Science Center at San Antonio Internal Research Board, protocol number 20220246EX.

Sample survey: The Quick Environmental Exposure Survey Instrument (QEESI) has emerged as a well-validated, widely used instrument for assessing CI [2]. To date, researchers and clinicians in sixteen countries have used the QEESI, which offers high sensitivity and specificity for differentiating individuals with CI from the general population [48,49] (see Palmer et al., [19] for a recent listing of global studies using the QEESI. The QEESI is available at https://tiltresearch.org/self-assessment/) (assessed on 1 December 2025).

Respondents completed two scales (the Chemical Intolerance and Symptom Severity scales) of the Quick Environmental Exposure and Sensitivity Inventory (QEESI)—a 50-item, internationally validated questionnaire designed to assess the symptoms and chemical intolerances of various chemicals, food, and drug exposures.

The QEESI consists of 4 scales and one index (Chemical Sensitivity, Symptom Severity, Life Impact, Other Intolerances, Masking Index), but only the Chemical Sensitivity and Symptom Severity scales are typically used to classify CI. The other scales are typically relevant in clinical practice. Each scale contains 10 items that are rated from 0 to 10 on a Likert scale: 0 = “not at all a problem” to 10 = “severe/disabling symptoms”. Total scores for each scale range from 0 to 100. Cut-off criteria for ‘High CI’ were scores ≥ 40 on both the Chemical Exposures and Symptom Severity scales. This is regarded as “very suggestive” of CI. Scores from 20 to 39 on one or both scales are considered “Medium or mid-level CI.” Scores less than 20 on both scales are classified as “Low CI”. These groupings were used for data analysis.

All respondents were asked two questions concerning their children: (1) “Has a doctor or health professional every told you that your child had Autism, Asperger’s disorder, pervasive developmental disorder, or autism spectrum disorder?” and (2) “Has a doctor or health professional every told your that your child has Attention Deficit Hyperactive Disorder or similar learning disability?” Of course, this applied to only those with children (*N* = 3172), and the number of children in the family was not assessed in this survey. These specific ASD/ADHD questions have been used by the National Institute of Health in other population surveys. Notwithstanding, it is emphasized here that these are parental self-reports regarding their assessment of any one of their children and are not clinically verified diagnoses and are used for associational statistical analysis purposes only, not epidemiological estimates.

Data Quality Control Checks: The 5000 survey records were assessed for data quality (DQ) encompassing completeness, validity, or accuracy concerns; four measures were required to exclude all surveys indicating one or more DQ concerns. Records with these concerns were excluded from the analytic data set. Table 1 presents the data quality exclusions by country. The United States sample had the largest data quality exclusions, whereas Mexico had the fewest.

Analysis of the DQ-flagged surveys were shown to be biased toward younger, U.S. males. The DQ-removed data set is 1.2% more female than the complete data set (ranging from 0.27% more female in Italy to 2.6% more female in India). The DQ-removed data set is 1.9% older (larger portion of the 55 and older group). U.S. skewed heavily older, with a 5.2% older population, and Mexico was only 0.46% older. Overall, the percentage of reporting children did not change much between the DQ-removed data set and complete data set. However, India reported having 1.16% less children, while Japan reported 2.2% more reporting children. Although we recognize the potential biases of excluding DQ-flagged surveys, we believe there is greater accuracy by excluding the problematic responses rather than using the whole sample. We have taken this approach to help improve some well-known DQ concerns associated with web-based surveys, including response probabilities and biases [50,51]. Stability and sensitivity analyses with and without the DQ-flagged surveys show negligible impact on the model’s results.

Figure 1 depicts the flow of data exclusions leading to the final analytic data set. Some of the DQ measures might technically be accurate (e.g., “male and breast implants”), but with an abundance of caution, they were excluded. The same could be said for the survey time measure: with a survey containing 40 questions, it is unlikely that a respondent could read and respond accurately to all questions in under two minutes. By omitting any records that violated one or more DQ measures, 604 records were excluded (12.1%)

QEESI Reliability by Country: Internal consistency of the two QEESI scales (Chemical Intolerance and Symptom Severity) was assessed using Cronbach’s alpha, calculated by country to evaluate scale reliability across cultural contexts.

Statistical Modeling: Although quasi-separation was not identified, low cell sizes for ASD and ADHD were identified by country in the three-level CI grouping in the Low and Medium categories. To alleviate this, CI was used as a binary variable: High CI (coded 1) combined with the Low and Mid CI classes (coded 0). This creates a “High CI” category and a “Low CI” Category with reasonable cell sizes.

A generalized linear model with a binomial distribution and logit link was used to estimate the association between High CI and ASD and ADHD. Because ASD and ADHD have a high comorbidity potential, they were run as outcomes in separate models and, therefore, were not adjusted for each other to avoid confounding. Respondents who reported having no children were excluded from the model. The model follows the functional form in Equation (1):Logit (p|x) = α + β1X1 + βiXi… + e(1)
where Logit(p) = log (P/1−p) represents the odds of y = 1 (ASD or ADHD present) versus y = 0 (condition absent). CI class (High vs. Low) served as the primary binary predictor X1. Additional covariates Xi included age category and sex. Income was not measured due to difficulties in harmonizing an appropriate SES level across countries. ASD and ADHD were used as outcomes in separate models. Given the potential for small-sample bias or separation in logistic models with categorical predictors, all models were estimated using Firth’s penalized likelihood method, which applies a bias-reducing Jeffreys-prior penalty to the score function [52,53]. This approach produces finite and more stable parameter estimates in the presence of spare cells or quasi-complete separation. Firth-adjusted estimates were obtained using the Bias-Adjusted parameterization in JMP 18.2 (SAS Institute) [54].

To detect an Odds Ratio of at least 2.0, assume that α = 0.05, 95% power, and that other covariates contribute r^2^ = 0.2; *N* = 225 is required. With an analytic *N* = 4396, this study is well-powered to detect smaller effect sizes, notwithstanding some data cells may be too sparse to claim this kind of power.

## 3. Results

Sample age and sex: For the total sample, Table 2 shows that age and sex categories were reasonably evenly distributed. There were no sex distribution differences across countries. There were, however, significant age distribution differences across countries. India and Mexico reported a greater percentage of younger respondents, and a fewer percentage of those over 65 years old compared to Italy, Japan, and the U.S. Further associated with age was having a child with ASD or ADHD. Reporting to have a child with ASD or ADHD decreased rapidly as age categories increased—with a distinct linear association for India and the U.S (see Appendix A for ASD and ADHD graphs, respectively). This age-group effect was evident across all countries except Japan.

QEESI Reliability by Country: The reliability of the QEESI was first presented by Miller and Prihoda (1999) [48,49]. In this study, both QEESI scales showed high reliability across all national samples (Table 3). Cronbach’s alpha values for the Chemical Intolerance scale ranged from 0.92 (Mexico) to 0.94 (Japan, USA), and alphas for the Symptom Severity scale ranged from 0.90 (Mexico) to 0.95 (India). These values are similar to previous studies in the USA, Japan [55], Sweden [56] and Denmark [57], and indicate good internal consistency across countries.

Chemical Intolerance: Figure 2 depicts the distribution of CI across countries. Across all countries, most respondents reported Mid or High CI levels compared to those for Low CI. Across all countries, the reported percentage of High CI is greater than that of Low CI with wider discrepancies in India, Japan, and Italy compared to Mexico and the U.S.

ASD/ADHD: Figure 3 and Table 4 depict the rates of families reporting at least one child with ASD and ADHD for each country by CI category. Despite a wide range of differing rates between groups, the reported rates of ASD and ADHD show similar patterns across all countries. Those in the highest CI category report higher ASD and ADHD rates than those in the Mid group, which in turn show higher rates than the Low CI group (see Figure 3 and Table 4). This pattern was most pronounced in India, the United States, and Mexico.

Table 5 shows the Odds Ratios (OR) and 95% Confidence Intervals of reporting at least one child with ASD and ADHD as a function of CI by country. The pattern of association between CI category and reporting at least one child with ASD and ADHD are similar. Compared to the Low CI category, the High CI category has significantly greater odds of reporting a child with ASD and ADHD. This was observed in all countries except Japan. Generally, sex is not a significant predictor, and younger-aged respondents have higher odds of reporting at least one child with ASD or ADHD (India, Italy, and the USA). Mexico was the only country showing a significant association between female respondents and reporting at least one child with ASD. The female demographic was not significant in any of the ADHD models.

Due to the high reporting of at least one child with ASD/ADHD in India, we conducted a sensitivity analysis by removing India from the data set. The sensitivity analysis did not change the *p*-values for the High CI estimates (*p* < 0.0001) and only negligibly lowered OR estimates (about −0.2 in ASD, and −0.1 in ADHD).

## 4. Discussion

The reported rates of CI, ASD, and ADHD varied considerably across countries, and it important to understand that these data cannot be used to estimate actual prevalence. The many methodological and cultural reasons for this are discussed in more depth in the limitations section below. Foremost, due to a failure to collect family size (number of children), the unit of measure of ASD/ADHD is at the family level. This measure is the per family incidence of ASD/ADHD. Secondly, ASD/ADHD were not clinically verified but based solely on parental self-report.

Self-reporting of CI and children’s ASD/ADHD in the same survey introduces a potential shared method variance and may influence a general tendency to endorse health and environmental problems. This could plausibly account for some of the observed associations.

Notwithstanding, the primary focus of these analyses is the consistent pattern between the degree of CI and its association with ASD/ADHD. In all countries, the High CI severity group had substantially higher odds of reporting a child with ASD/ADHD than the Low CI severity group. These findings are consistent with two prior U.S. studies showing a similar significant association between ASD/ADHD and CI [46,47]. Nevertheless, the associational studies cannot be interpreted as causal associations until proper longitudinal studies are completed.

Study Limitations:

1. We relied on parental self-reports to assess ASD, ADHD and CI. Verified diagnoses should be conducted through expert clinical behavioral observation and interview assessment. It is important to consider this limitation and that the rates reported in this study are family-level rates (that is, at least one child is reported as having ASD/ADHD). This family rate is not comparable to global prevalence estimates (1–2%).

2. Two prior U.S. studies with different survey methodologies demonstrated that those with CI had a 3-fold increased probability for reporting a child with ASD/ADHD. However, the methodologies are not directly comparable. While the current study is the first to demonstrate an association in four other countries, there is no other evidence in the literature to compare this association in other countries with similar methods.

3. It is also likely that other unmeasured confounding factors that influence CI itself contribute to the associations we found with ASD/ADHD in this study. Perhaps the most prominent likely confounder in this study is the failure to adjust for socioeconomic status (SES or education). This exclusion is problematic because it is well-documented that differing levels of SES can affect CI and ASD diagnosis, exposure to environmental chemicals, chronic stress and access to healthcare. The omission of collecting SES and education in our analysis has most likely affected our results. Our data cannot address any effects of a potential spurious relationship with SES. While our results are consistent with prior studies, this study should be considered as another preliminary step in understanding the association between CI and ASD/ADHD

4. The parent-reported family-level ASD rates in this study far exceed global prevalence estimates (1–2%), raising concerns about survey validity. The survey was self-reported for both CI and maternal reports of child ASD/ADHD. This shared method variance could artificially inflate our found association. This could introduce response-style biases due to personality traits, or a respondent’s idea about the intended relationship between variables; for example, consistently using extreme ends of a response scale for all questions or responding to questions based on mood, health anxiety, or hypersensitivities rather than the responding to the specific content of the question. 

5. Reliance on an online platform for recruitment is likely to contribute to sampling bias by excluding populations with low education levels or limited digital access. This could undermine the generalizability of findings, particularly in low-resource settings. As described in the Data Quality Control Checks section, precautions were taken to mitigate known data-quality concerns associated with web-based surveys, including response probability and related biases [50,51].

Further, the use of online invitations to participate (emails, phone alerts, online banners, and community-site messaging) rather than pulling from established research panels could lead to a biased pool of respondents who are interested in the subject area of research (CI/ASD/ADHD). As such, respondent interest could have influenced some portions of the family-level reports. Future studies should strongly consider the impact of this influence.

6. Cultural heterogeneity: Back-translation of the survey was performed; however, it is unknown how the distinct cultural norms and symptom-reporting practices in each country affect the results. Cultural validation procedures such as cognitive interviewing were not conducted. For example, Japan’s data reflects specific cultural sampling biases, including a likely social stigma reporting bias. Cross-cultural differences in survey response styles show that Asians tend to respond more modestly and demonstrate more middle-level items on Likert scales then Westerners [58,59,60,61] (as seen in the Appendix A).

This assertion was examined in our data using the QEESI item responses. Indeed, we found Japan’s responses to be consistent with Wang et al. [61], where respondents from Japan stand out as choosing middle responses. This cultural phenomenon is depicted in Appendix A.

Similarly, among all countries in this study, other unknown cross-cultural reporting biases, sociocultural diagnostic differences, differential base rates, and other measurement artifacts severely limit cross-cultural comparison. Therefore, because ASD prevalence differs drastically by nation, Odds Ratios are not directly comparable without reference to a baseline. Given the limitations, no causal interpretation can be inferred, and cross-cultural generalizability of this study should be conducted cautiously.

7. The cross-sectional design prevents distinguishing whether CI precedes or co-occurs with the recognition of a child’s diagnosis. These confounders and potential interactions have yet to be understood and could be explored in future studies.

Future Directions: Given these limitations presented above, the results of this study cannot be used to understand the biomechanics, directionality of the association, or to directly inform public policy. This will require future studies to utilize carefully planned experimental or clinical trials, rather than survey-based assessments, as in this study. Understanding the causation between the association of CI and ASD/ADHD would require a deeper understanding of the shared biological mechanisms involved.

As such, future studies might investigate plausible multi-omics biomechanisms involving the complexity of gene–environment interactions (G × E) and epigenetic modulation [62], which are now regarded as the most probable explanation for most idiopathic ASD cases [63,64,65]. This is especially pertinent since known autism genes selectively target metabolic clearance proficiency for pesticides, heavy metals, bisphenol A, and phthalates, which are already known as endocrine disruptors and neurotoxins [66]. The biological risks of toxic exposure profiles are multifaceted and synergistic, involving oxidative stress, mitochondrial dysfunction, and systemic immune activation [67,68].

The specific environmental and biological mechanisms underlying ASD, ADHD, and CI remain poorly understood and likely vary substantially across individuals. Therefore, future studies would need to focus on a deeper understanding of the biologically mediated effects of toxic exposures. These kinds of studies would be needed to properly inform public policy or guide regulation of environmental toxicants [69,70].

Our study did not assess indoor or outdoor air quality; nevertheless, based on the broader literature, we feel it important to mention that one potential area for future investigations may involve outdoor air quality. Systematic review studies confirm an association between outdoor air pollution and ASD [71,72,73,74]. Relevant to this, we conducted a post hoc analysis where we investigated the idea that countries with high degrees of air pollution would have higher rates of ASD compared to countries with lower air pollution.

Inspection of Figure 4 shows that India is among the countries with the worst air quality and highest autism rates. India also had the highest CI rates in this study. Interesting, Italy and Japan, which did not report as large of a percentage between CI and ASD as the other countries, ranked among the countries with better air quality and lower rates of ASD. Certainly, interpretation of this ecological data cannot be extrapolated to the individual level [75]. Notwithstanding, based upon the broader air quality and health literature, the association between CI, ASD/ADHD, and air quality (both indoor and outdoor) warrants further investigation to inform recommendations for public health policy [69,70].

Poor indoor air quality (IAQ) may also be considered for future studies. Poor IAQ can lead to oxidative stress and inflammation processes related to neurological and fetal health [76,77,78,79,80,81]. The sources of poor IAQ (PM2.5) are volatile organic compound sources that include cooking with gas, pesticides, personal care/cleaning products, chemicals from new construction materials/furniture, mold, use of incense, candles, and pet allergens. Two recent studies have demonstrated that interventions designed to reduce symptom severity among participants with CI were successful [82,83].

## 5. Conclusions

This international study is consistent with two prior U.S. studies that also demonstrate increased reported ASD/ADHD rates reported among those with High CI. This similar pattern observed over the countries is the takeaway message of this study. The widely varying rates of CI, ASD, and ADHD by country should not obfuscate this connection between CI and ASD/ADHD. As noted above in the limitations section, there are various methodological issues that prevent us from drawing any conclusions about cross-country prevalence rate comparisons.

This association between CI and ASD/ADHD is not fully understood. There remain key unanswered questions concerning the biological mediating or moderating mechanisms. There are many biological pathways that differ across individuals. During pregnancy and early childhood, all developing organ systems may be vulnerable to a variety of toxic exposures. Though specific mechanisms are unknown, based on the broader literature, a precautionary principle may be used as a first intervention step. Whether a mother has CI or not, expecting parents may be advised to reduce toxicant exposure by identifying and limiting household xenobiotics such as chemicals in the air, food, drugs, water, or construction materials that may potentially trigger biological processes underlying CI and ASD/ADHD.

## Figures and Tables

**Figure 1 jox-16-00005-f001:**
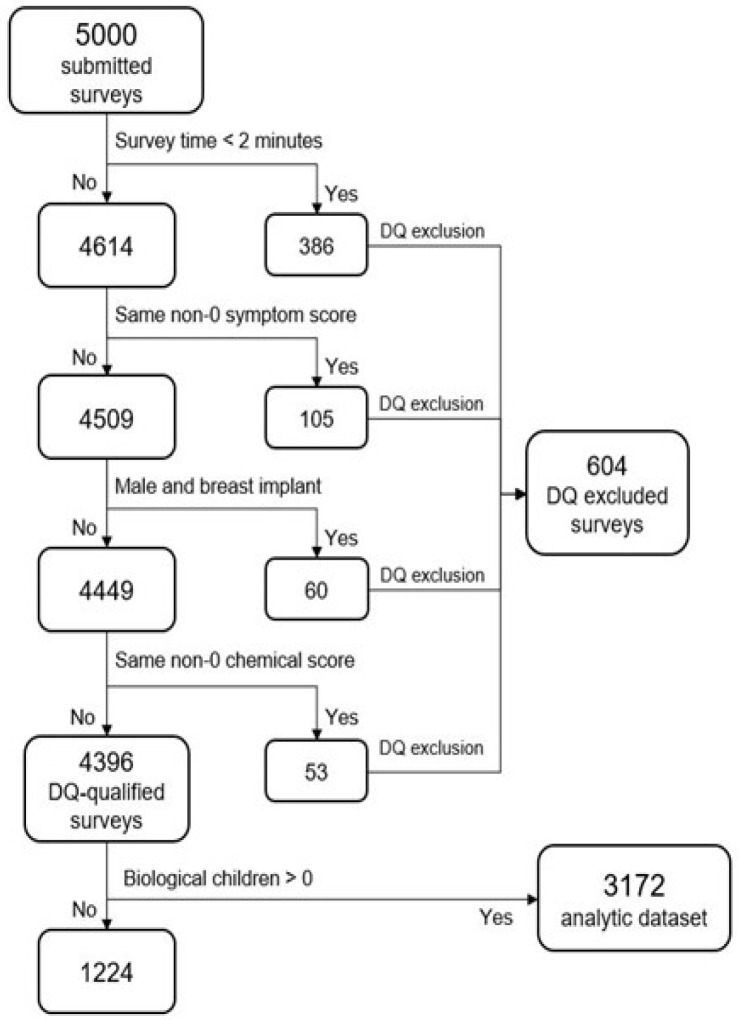
Data flow exclusions.

**Figure 2 jox-16-00005-f002:**
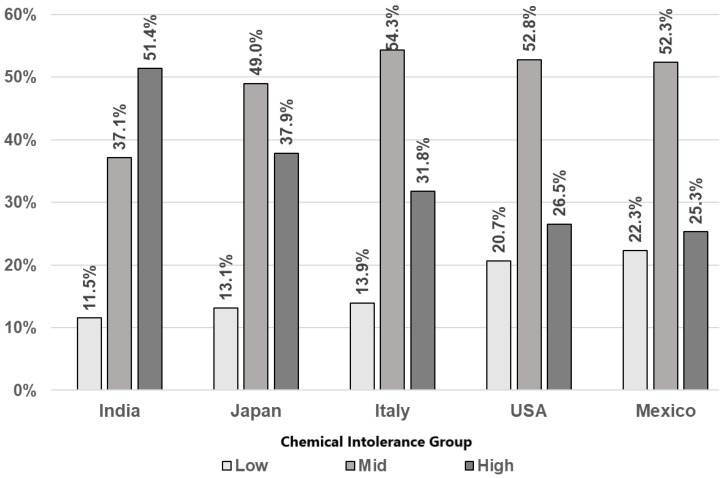
Rates of Chemical Intolerance class by country.

**Figure 3 jox-16-00005-f003:**
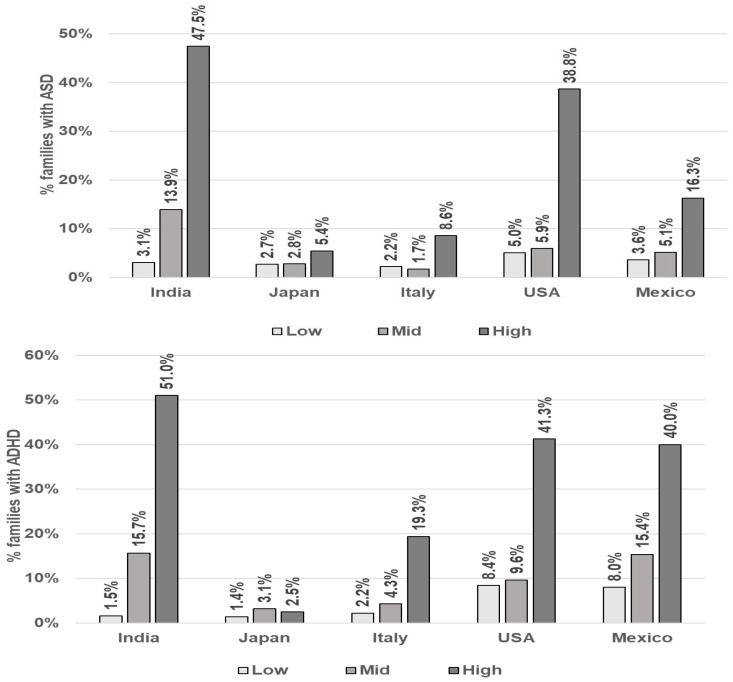
Rates of parental self-reporting of at least one child with ASD and ADHD by Chemical Intolerance Class and Country.

**Figure 4 jox-16-00005-f004:**
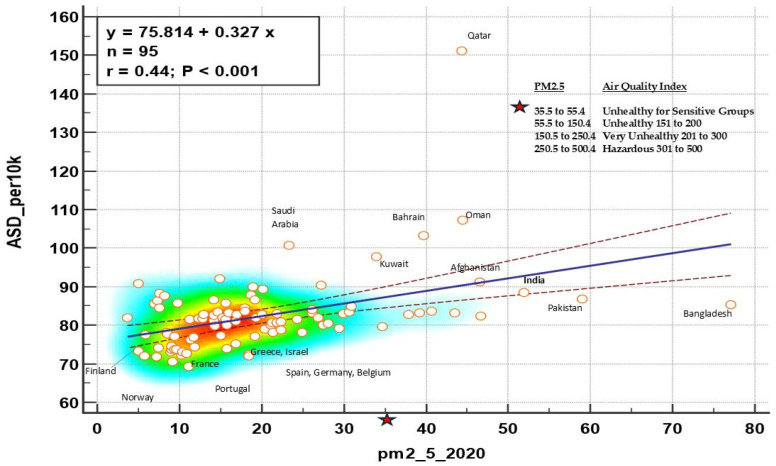
Global rates of ASD by PM2.5 Air Quality Index. Lines correspond to the regression line with 95% confidence intervals, circles are countries,

**Table 1 jox-16-00005-t001:** Data quality (DQ) exclusions by country.

Country	Exclusions	Percent
USA	212/1000	21.2%
Japan	155/1000	15.5%
Italy	103/1000	10.3%
India	97/1000	9.7%
Mexico	37/1000	3.7%

**Table 2 jox-16-00005-t002:** Sample demographics across countries.

		Total Sample Percent	India	Japan	Italy	USA	Mexico	*p* ChiSq
Age	18 to 34	32.2%	44.2%	24.7%	21.5%	25.0%	43.2%	*p* < 0.0001
	35 to 54	35.9%	38.3%	30.9%	36.9%	36.8%	36.3%	
	55+	31.9%	17.5%	44.4%	41.6%	38.2%	20.5%	
Sex	Male	47.7%	48.8%	47.5%	47.8%	46.2%	47.8%	*p* < 0.0537
	Female	52.2%	51.2%	52.4%	52.2%	53.2%	52.1%	
CI Group	Low	16.3%	11.5%	13.1%	13.9%	20.7%	22.3%	*p* < 0.0001
	Medium	49.0%	37.1%	49.0%	54.3%	52.8%	52.3%	
	High	34.6%	51.4%	37.9%	31.8%	26.5%	25.3%	
Children	Yes	72.2%	76.7%	63.0%	75.1%	69.7%	75.2%	*p* < 0.0001
	No	27.8%	23.3%	37.0%	24.9%	30.3%	24.8%	
ASD ^a^	Yes	13.1%	32.2%	3.8%	4.2%	15.3%	8.2%	*p* < 0.0001
	No	87.0%	67.8%	96.2%	95.9%	84.7%	91.9%	
ADHD ^a^	Yes	18.0%	34.6%	2.6%	9.2%	18.6%	21.3%	*p* < 0.0001
	No	82.0%	65.4%	97.4%	90.8%	81.4%	78.7%	

^a^ These are parental self-reports at the family level; having at least one child reported having the condition. This does not imply clinical diagnosis. These rates exceed epidemiological rates and should not be interpreted as prevalence estimates or compared across countries due to biases in the survey methods.

**Table 3 jox-16-00005-t003:** Cronbach’s alpha scores for the Chemical Intolerance and Symptom Severity scales by country.

Country	Cronbach’s Alpha	
	QEESI Scale	
	Chemical Intolerance	Symptom Severity
India	0.93	0.95
Japan	0.94	0.94
Italy	0.93	0.92
USA	0.94	0.92
Mexico	0.92	0.90

**Table 4 jox-16-00005-t004:** Rates of maternal self-reports reporting of at least one child with ASD and ADHD by Chemical Intolerance class and country.

	^a^ Parental Self-Reported Family-Level ASD										
		India		Japan		Italy		Mexico		USA	
		N	% *	N	% *	N	% *	N	% *	N	% *
**High**	*yes*	189	47.5%	11	5.4%	20	8.6%	35	16.3%	62	38.7%
	*no*	209		193		213		180		98	
**Mid**	*yes*	32	13.9%	7	2.8%	6	1.7%	19	5.1%	16	5.9%
	*no*	198		247		345		352		254	
**Low**	*yes*	2	3.1%	2	2.7%	2	2.2%	5	3.6%	6	5.0%
	*no*	63		72		88		133		113	
**CI Class**	**^a^** **Parental Self-Reported Family-Level ADHD**										
**High**	*yes*	203	51.0%	5	0.1%	45	19.3%	86	40.0%	66	41.3%
	*no*	195		199		188		129		94	
**Mid**	*yes*	36	15.6%	8	3.1%	15	4.3%	57	15.4%	26	9.6%
	*no*	194		246		336		314		244	
**Low**	*yes*	1	1.5%	1	1.3%	2	2.2%	11	8.0%	10	8.4%
	*no*	64		73		88		127		109	

^a^ This is reported at the family level with at least one child reported to have the condition. ASD/ADHD were not clinically diagnosed in this study. The rates exceed epidemiological rates and should not be interpreted as prevalence estimates or compared across countries due to biases in the survey methods. * Percentage calculated by dividing the yes responses in each class and dividing by the total N in that class. These are the percentages reported in Figure 3.

**Table 5 jox-16-00005-t005:** Odds Ratios for CI predicting parental self-reported family-level ASD and ADHD by country.

Effect	Country				
	India (*N =* 693) *	Japan *N* = 532 *	Italy *N* = 674 *	Mexico *N* = 724 *	USA *N* = 549 *
	OR (95% CI)	OR (95% CI)	OR (95% CI)	OR (95% CI)	OR (95% CI)
**ASD**					
High vs. Low CI ^a^	**2.5 (2.0–3.0) *****	1.4 (0.9–2.2) ^ns^	**2.0 (1.3–3.0) ****	**1.9 (1.5–2.5) *****	**2.7 (2.1–3.6) *****
Age Group	**0.5 (0.4–0.6) *****	0.6 (0.3–1.0) ^ns^	**0.4 (0.3–0.8) ****	0.7 (0.5–1.0) ^ns^	**0.3 (0.2–0.5) *****
Female	1.2 (1.0–1.4) ^ns^	1.5 (0.9–2.4) ^ns^	0.8 (0.5–1.2) ^ns^	**0.7 (0.5–0.9) ***	0.9 (0.7–1.2) ^ns^
**ADHD**					
High vs. Low CI ^a^	**2.5 (2.1–3.1) *****	0.9 (0.5–1.6) ^ns^	**2.3 (1.7–3.1) *****	**2.0 (1.7–2.5) *****	**2.3 (1.8–2.9) *****
Age Group	**0.5 (0.4–0.6) *****	0.6 (0.3–1.2) ^ns^	**0.7 (0.5–1.0) ***	0.8 (0.6–1.1) ^ns^	**0.5 (0.4–0.7) *****
Female	1.2 (1.0–1.4) ^ns^	1.3 (0.7–2.2) ^ns^	1.0 (0.8 –1.3) ^ns^	1.0 (0.8–1.1) ^ns^	1.1 (0.9–1.4) ^ns^

Bolded numbers are statistically significant: ns = not significant, * *p*-value less than 0.05, ** *p*-value less than 0.01, *** *p*-value less than 0.001. ^a^ low CI = combined Low and Mid CI due to low cells for ASD/ADHD in each. See Table 4.

## Data Availability

The data presented in this study are openly available in the UTHSCSA data repository at https://dataverse.tdl.org/dataverse/ci (accessed on 1 December 2025).

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
