# Peer review of "Chemical Intolerance Is Associated with Autism Spectrum and Attention Deficit Disorders: A Five-Country Cross-National Replication Analysis"

_jox, 2026, doi:10.3390/jox16010005_

Round 1
Reviewer 1 Report
Comments and Suggestions for Authors
This study is very interesting as it is the first to reveal that, despite the large differences in the prevalence of chemical intolerance (CI) and autism spectrum disorder (ASD) across five countries (United States, India, Italy, Japan, and Mexico), children of parents with high CI scores have a higher prevalence of ASD was consistent across all four countries except Japan. However, the above content is difficult for readers to understand because it is a little insufficient in explanation. Therefore, the following additions and corrections should be addressed.
1) Were respondents also asked about the prevalence of ADS and ADHD? Or were they only asked about the prevalence of their children? Please clarify this point.
2) Line 108: Respondents answered a 40-item survey. Please specify the 40 questions (as an additional table) to help readers understand the content of the questions.
3) Please provide an explanation of the words under all tables and figures.
For example, in Table 2, does ASD and ADHD represent the prevalence among individuals or their children? What is QEESI?
4) Line 277: There is no figure 5. Should it be Table 5?

Author Response
Word Doc attached

Reviewer 2 Report
Comments and Suggestions for Authors
The idea of exploring CI as a potential proxy for environmental vulnerability is interesting and aligns with broader discussions of gene–environment interactions, and neurodevelopmental outcomes. However, the study’s methodological and analytical choices introduce significant limitations that affect the strength and interpretability of the findings.
Abstract:
In the abstract, the prevalence values (e.g., 32% ASD in India) are implausibly high by international epidemiological standards. They should be clearly acknowledged as self-reported rather than clinically confirmed.
Why was the word validation chosen in the title when the methodology cannot verify diagnostic accuracy?
Replace “validation” with “exploratory cross-national analysis”.
Introduction:
In the introduction, why is there minimal discussion of controversies surrounding CI (diagnostic heterogeneity, psychological vs physiological debate)?
Can the authors clarify the biological rationale linking adult CI to child neurodevelopment without suggesting causality?
Why is prior U.S. research presented without attention to its own limitations (self-report, cross-sectional)?
The narrative sometimes drifts into a general mini-review on environmental risk factors in ASD/ADHD, rather than staying tightly anchored to CI.
State clearly: “parent-reported ASD/ADHD diagnoses” rather than “ASD” or “ADHD”.
Tone down causal language: use “associated with”, not “risk”.
Methodology:
A cross-sectional online survey cannot validate medical diagnoses. The sampling frame (Dynata panel) is non-probabilistic and may introduce strong selection bias.
The lack of socioeconomic variables removes a major confounder influencing both CI and ASD diagnosis.
Why were SES variables omitted when they are known predictors of ASD diagnosis and chemical exposure?
Were respondents asked about the number of children? If not, how is the ASD/ADHD risk per “child” interpreted?
What were the translation validation procedures? Were cognitive interviews conducted?
Measuring ASD and ADHD via a single yes/no parental question is weak, especially across cultures with varying diagnostic practices. The reported rates (2.6–34%) do not align with global data.
Please explicitly state that child diagnoses were not clinically validated.
Could you highlight in both Methods and Limitations that prevalence estimates are unreliable?
QEESI is a validated tool, but its cultural transferability requires reporting of psychometrics per country (Cronbach’s alpha, consistency). This is missing.
Did the authors check the internal reliability of QEESI by country?
Were cutoffs validated cross-culturally or carried over directly from U.S. data?
Excluding 12.1% of the data, with markedly different exclusion rates across countries (USA: 21% vs. Mexico: 3.7%), may introduce systematic bias. Why are exclusion rates so uneven across nations? Could “illogical” responses stem from cultural or linguistic misunderstandings rather than poor data quality? Justify the exclusion criteria. Examine whether exclusion disproportionately removed specific demographic groups.
Statistical analysis
The models adjust for age and sex but omit key confounders (income, education, and healthcare access). The massive ORs (e.g., 51.9 in India) signal sparse data problems. There is no correction for multiple comparisons.
Were models tested for separation or instability?
Were penalised logistic methods considered (e.g., Firth regression)?
Why were SES variables excluded despite their importance?
I suggest applying penalised regression, adding SES where possible, or acknowledging its absence more strongly, and Tone down the emphasis on the magnitude of ORs; focus instead on pattern consistency.
Results
Results are clearly presented, but the prevalence values undermine interpretability. The linear trend between the CI category and ASD/ADHD is consistent but likely driven by reporting tendencies.
I suggest adding a sensitivity analysis excluding extreme prevalence values, and consider adjusting for the number and age of children per respondent.
Discussion
Move the mechanistic content to a short future-directions section.
Expand discussion on shared reporting tendencies (e.g., health anxiety).
Why include ecological comparisons that do not involve individual exposure data?
Could this section be removed or minimised?
Start with the limitations of diagnostic validity concerns.
Clearly state: “These data cannot be used to estimate ASD/ADHD prevalence.”
In conclusion, replace “replication” with “similar pattern observed” and add explicit caution about causal inference.
References
Comprehensive, but over-reliant on literature linking environmental toxins to ASD. Very few citations reflect scepticism about CI or diagnostic variability.
Author Response
Word file attached

Reviewer 3 Report
Comments and Suggestions for Authors
This manuscript titled ‘Chemical intolerance and autism: a five-country validation study’ by Palmer and Kattari, reports a five-country web-based survey (N = 4,396 of qualified surveys) examining associations between chemical intolerance (CI), measured with QEESI, and parent-reported diagnoses of ASD and ADHD in their children. Across most countries, higher CI categories are associated with higher odds of reporting a child with ASD and/or ADHD. The topic is important, and the international replication effort is valuable. However, there are substantial concerns related to measurement, bias, and interpretation. In my view, these limitations preclude publication in its present form.
Major Comments
- The foremost concern is the validity of ASD/ADHD measures and implausible prevalence. In this study, the ASD/ADHD are assessed via single parent-report items. The resulting prevalence estimates are far above accepted epidemiologic ranges in several countries (e.g., >30% for ASD/ADHD in some strata like USA and India), strongly suggesting substantial misclassification and/or misunderstanding of the questions. The manuscript currently treats these as if they were epidemiologic rates, which is not justified. This limitation needs to be emphasized prominently in the Abstract, Results, and Discussion, and conclusions must be tempered accordingly.
- Similarly, in this study, CI and child ASD/ADHD are both self-reported in the same survey, likely subject to shared method variance and a general tendency to endorse health and environmental problems. This could plausibly account for much of the observed association. This alternative explanation should be explicitly discussed, and, where possible, sensitivity analyses (e.g., adjusting for general symptom or health-complaint indices) should be attempted or at least proposed.
- The cross-sectional, retrospective design does not establish whether CI precedes the child’s diagnosis. CI could arise after the stress of caring for a neurodivergent child, or both could be linked to third variables. The manuscript sometimes implies that CI is a risk factor for ASD/ADHD; this should be consistently reframed as a correlational finding. Causal or mechanistic language (e.g., CI “contributing to” ASD/ADHD) should be softened throughout, including in the Abstract and conclusions.
- Sampling, representativeness, and between-country comparisons
The online panel is not nationally representative, and data-quality exclusions vary markedly by country. This likely biases prevalence and may affect cross-country comparisons. The authors should 1) clearly state that the sample is not suitable for estimating population rates, 2) clarify whether analyses were weighted, and 3) more explicitly discuss how panel recruitment and exclusion patterns may limit generalizability. - Some odds ratios are extremely large with very wide confidence intervals, reflecting sparse cells (especially in Low CI groups). These estimates are unstable and should be clearly flagged and interpreted with great caution. Collapsing categories or moving the more granular country-specific estimates to supplementary material might improve robustness. Clarification of how age was coded and modeled is also needed, as is an explicit acknowledgment that ASD and ADHD comorbidity is not modeled (e.g., ASD models do not adjust for ADHD and vice versa).
- The Discussion covers extensive mechanistic hypotheses (gene–environment interactions, mitochondrial dysfunction, mast cells, air pollution), but none of these pathways are measured in the present data set. This section should be tightened and clearly marked as speculative and hypothesis-generating, clearly separated from the empirical findings.
Minor Comments
- The Abstract should explicitly state that ASD/ADHD diagnoses are parent-reported from a web survey, and that observed prevalence exceeds known population estimates.
- Methods should clarify data-collection dates, handling of participants without children, and whether panel weights were used.
- Tables and figures would benefit from improved formatting and clearer legends (especially definitions of CI categories).
- Policy/clinical recommendations about advising expecting parents to reduce toxicant exposure should be clearly grounded in the broader literature, not attributed to this survey alone.
Author Response
Word file attached

Round 2
Reviewer 2 Report
Comments and Suggestions for Authors
Dear Authors,
Thank you for the careful and extensive revisions you have made. Many of the earlier concerns have been addressed, and the manuscript is now clearer in several key areas, particularly the clarification of self-reported child diagnoses, the addition of country-level QEESI reliability statistics, and the implementation of Firth-adjusted models. These additions strengthen the methodological transparency and improve the credibility of the overall analysis.
That said, some important issues remain either partially resolved or insufficiently elaborated, and further refinement is still needed before the manuscript presents a fully balanced and methodologically grounded interpretation of the findings.
1. Limited discussion of CI diagnostic complexities: The introduction still offers only a brief acknowledgement of the long-standing controversy surrounding CI, particularly regarding diagnostic heterogeneity and the psychogenic vs. physiological debate. Because CI is the central exposure variable, a fuller, more even-handed description of these issues is essential for readers who may not be familiar with the field.
2. Influence of unmeasured SES factors: Although the limitations section notes the absence of SES variables, the implications of this omission require more precise articulation. Given well-documented SES gradients in ASD diagnosis, chemical exposure, and access to care, the lack of socioeconomic adjustment likely has a meaningful impact on the observed associations and should be addressed more prominently.
3. Data exclusion and potential demographic skew: While the data-quality exclusions are described in detail, the manuscript does not explore whether these exclusions disproportionately affected particular demographic subsets within each country. Considering the significant differences in exclusion rates across nations, even a brief descriptive comparison would help reassure readers that the exclusions did not introduce new systematic biases.
4. Sensitivity analyses: The added sensitivity analysis using the whole dataset is helpful; however, it does not address the reviewer’s original suggestion to conduct a sensitivity analysis excluding countries with extreme prevalence values (e.g., India). Even a simplified version, such as re-running the primary model without the highest-prevalence country, would give readers a clearer sense of the stability of the CI–ASD/ADHD association.
5. Cultural and linguistic validation: Back-translation is mentioned, but there remains no discussion of cultural validation procedures such as cognitive interviewing. Because the QEESI is being used across five countries with distinct cultural norms and symptom-reporting practices, even a brief acknowledgement of the absence of cultural validation would help contextualise interpretive limits.
6. Reporting tendencies and shared method variance: The manuscript now includes a brief comment on shared reporting tendencies, but this section would benefit from slightly more detail. Readers would be better served by a more precise explanation of how general health-symptom endorsement patterns, health anxiety, or hypersensitivity to environmental issues might influence both CI scores and the likelihood of reporting a child diagnosis.
7. Scope of the ecological air-quality section: The ecological comparison in the discussion and future-directions sections is still quite extensive relative to its relevance to the main research question. Because no air-quality variables were measured in the study, a more concise version of this section may help maintain focus on what the data can reasonably support.
I appreciate the progress in this revised version and hope that the above suggestions help further strengthen the clarity, methodological balance, and interpretive caution of the manuscript.
Reviewer 3 Report
Comments and Suggestions for Authors
The authors substantially improved the manuscript. I have two remaining points listed below.
- Given that comorbidity between ASD and ADHD is common, I encourage the authors to add a statement in the Methods and/or Limitations, emphasizing that your models do not adjust ASD for ADHD (and vice versa), and that comorbidity was not explicitly modeled. This reinforces the already-strong caution about unmeasured confounding.
- I would encourage consistency check to ensure that throughout the abstract, tables, and main text, ASD/ADHD measures are always described as “parent-reported” or “family-reported” rather than as diagnoses, especially in headings and figure/table captions.
Author Response
please see atached
